# Prism adaptation treatment to address spatial neglect in an intensive rehabilitation program: A randomized pilot and feasibility trial

Tomas Vilimovsky[1]*, Peii Chen[2,3], Kristyna Hoidekrova[4,5,6], Jakub Petioky[5,7], Pavel Harsa[1]

1 Department of Psychiatry, First Faculty of Medicine, Charles University, Prague, Czech Republic,
2 Kessler Foundation, West Orange, NJ, United States of America, 3 Department of Physical Medicine and Rehabilitation, Rutgers University, Newark, NJ, United States of America, 4 Department of Rehabilitation Medicine, First Faculty of Medicine, Charles University, Prague, Czech Republic, 5 Rehabilitation Center Kladruby, Kladruby, Czech Republic, 6 Department of Kinanthropology, Faculty of Physical Education and Sport, Charles University, Prague, Czech Republic, 7 Third Faculty of Medicine, Charles University, Prague, Czech Republic

☯ These authors contributed equally to this work.
* tomas.vilimovsky@gmail.com

**Data Availability Statement:** All relevant data are within the manuscript and its Supporting Information files.

## Abstract

Spatial neglect (SN) is a common cognitive disorder after brain injury. Prism adaptation treatment (PAT) is one of the promising interventions for SN albeit inconsistent results from previous studies. We carried out a comparison intervention (PAT vs. Sham) and aimed to evaluate the efficacy of PAT on visuospatial symptoms of SN in an inpatient rehabilitation setting that offered a highly intensive comprehensive brain injury rehabilitation program. A total of 34 patients with moderate-to-severe SN secondary to stroke or traumatic brain injury were randomized to the PAT group and the Sham group (an active control group). Both groups received 10 sessions of treatment, over two weeks, in addition to the rehabilitation therapies provided by their rehabilitation care teams. Outcomes were measured using an ecological instrument (the Catherine Bergego Scale) and paper-and-pencil tests (the Bells Test, the Line Bisection Test and the Scene Copying Test). Patients were assessed at baseline, immediately after treatment, two weeks after treatment, and four weeks after treatment. 23 (67.6%) patients completed treatment and all the assessment sessions and were included in the final analyses using mixed linear modeling. While SN symptoms reduced in both groups, we found no difference between the two groups in the degree of improvement. In addition, the average SN recovery rates were 39.1% and 28.6% in the PAT and Sham groups, respectively, but this discrepancy did not reach statistical significance. Thus, the present study suggests that PAT may contribute little to SN care in the context of a highly intensive inpatient rehabilitation program. Further large-scale investigation is required to uncover the mechanisms underlying PAT and Sham in order to refine the treatment or create new interventions.

**Funding:** TV: This study was supported by the program PROGRES (Progres = C4 = 8D.Q 06/LF1 = 20). The program had no role in study design, data collection and analysis, decision to publish, or preparation of the manuscript.

**Competing interests:** I have read the journal's policy and the authors of this manuscript have the following competing interests: Peii Chen is employed by Kessler Foundation, receiving funding from the State of New Jersey and the US federal government. Her work related to the present study is, however, not funded by external funding agency. Peii Chen is listed as the inventor on the US Patent (No. 10,739,618) related to the key element of the treatment equipment reported in the manuscript. Other authors have declared that no competing interests exist. Following patent was used in this study: Wearable System and Methods of Treatment of a Neurocognitive Condition (U.S. Patent Number 10,739,618). Patent owner: Kessler Foundation. This does not alter our adherence to PLOS ONE policies on sharing data and materials.

## Introduction

Spatial neglect is characterized as a failure to report, respond or orient to stimuli presented in the side of space contralateral to the injured cerebral hemisphere, which cannot be explained by primary sensory or motor deficits [1]. The disorder is caused by damaged neural networks critical to spatial attention and related cognitive and motor functions, leading to a variety of symptoms [2–5]. While this neuropsychological syndrome has been mostly studied in stroke survivors, with the prevalence of 30–50% in acute-to-subacute stages [6–8], it can also be caused by other forms of brain pathology such as traumatic brain injury [9], neurodegenerative disease [10] or tumor resection [11]. Patients with spatial neglect tend to have poorer rehabilitation outcomes [8, 12–15], longer hospital stay [12–14, 16] and are less likely to be discharged home [16] than patients without spatial neglect. Spatial neglect, thus, predicts a decreased level of functional independence [7, 17, 18], increased use of health care resources [19] and increased family burden [7, 20].

One of the most promising and most commonly used interventions for spatial neglect is prism adaptation treatment (PAT) [21, 22]. Prism adaptation is a visuomotor phenomenon that had been known for decades [23] before it was used for treating spatial neglect [24]. During a prism adaptation session, patients wear goggles with prism lenses that shift the visual field horizontally to the ipsilesional side of space, and repeatedly perform arm-reaching visuomotor tasks that typically last less than 20 minutes. Upon subsequent removal of goggles, an after-effect can be observed as patients miss the target by reaching toward the contralesional side of space. After a few seconds to hours, the after-effect disappears. Prism adaptation and its after-effect occur effortlessly, requiring no explicit strategy learning. The phenomenon depends on cortico-cerebellar connectivity and involvement of neural networks critical to attention and sensorimotor integration [25–27]. That is, certain regions of the brain are activated at the time when prism adaptation occurs. Repeated sessions of prism adaptation may consolidate enhanced neural activation with strengthened brain connectivity among regions within and between ventral and dorsal attention networks that are impaired or dysfunctional in clinical populations, especially patients with spatial neglect. This, in turn, facilitates the restoration of spatial abilities that have been lost in those patients. Thus, prism adaptation procedures have been standardized into treatment protocols with multiple sessions over several days. Benefits lasting months to years of prism adaptation have been documented [28, 29] with positive effects not only on visuospatial abilities [24, 30, 31] but also on postural balance [32], motor functions [33], and activities of daily life (for a recent systematic review, see [34]).

PAT is recommended to rehabilitation care practitioners for treating spatial neglect by major practice guidelines in different regions of the world, such as the United States [35, 36], Australia [37], Canada [38], and the UK [39]. However, randomized controlled trials (RCTs) that implemented PAT within inpatient rehabilitation showed mixed results. Overall, studies that utilized prisms with weaker diopter (≤ 10 diopter; prism of 1 diopter shifts the visual field for approximately 0.57 degree) tend to result in negative findings [40, 41]. Even using prisms with stronger diopter (e.g., 20 diopter), results are inconsistent. For example, Mizuno et al. [42] found that compared to patients who received the treatment but wore flat lenses (i.e., Sham treatment), patients in the PAT group completing 20 sessions, two sessions a day, over two weeks showed no greater improvement in visuospatial ability (measured in a battery of paper-based tests) or visuospatial function (measured using an ecological assessment). In the same study, the authors found the PAT group, especially patients had relatively milder neglect at baseline, demonstrated greater improvement in rehabilitation outcomes at the end of inpatient stay than the Sham group [42]. In Vaes et al.'s study [43], after seven sessions over 7–12 days, the PAT group showed better outcomes than the Sham group in visuospatial ability

(measured using a battery of computerized tests). Ten Brink et al. [44], however, found that the extent of improvement in visuospatial ability (measured using a target cancellation test) and in visuospatial function (measured using an ecological assessment) did not differ between the PAT and Sham groups who completed 10 once-daily sessions over two weeks. It is important to note that in the studies reviewed above, patients receiving Sham treatment also showed improvement, but to what extent their improvement was in comparison to the PAT group differed across studies. Thus, it is questionable whether PAT or simply the visuomotor training without prism adaptation facilitates amelioration of spatial neglect during inpatient rehabilitation.

Ten Brink et al.'s study [44] was particularly informative as their sample size was larger than most of the RCTs published, and their description of the usual care indicated that, in addition to participating in the trial, patients received therapies addressing spatial neglect every day as part of their regular 4–6 therapy sessions. This may have accounted, to some extent, for their negative results of the RCT. Another possibility is that some of the patients participating in the study may have not been able to improve further after PAT given their relatively mild neglect at baseline. Patients with relatively severe neglect may have benefited from PAT to greater extents even in a setting that offered intensive rehabilitation services. Thus, in the present study of a randomized pilot and feasibility trial [45], we focused on patients with moderate to severe neglect. In order to closely represent the acquired brain injury population in inpatient rehabilitation settings, we included patients with left or right neglect secondary to stroke or traumatic brain injury (TBI). The study was conducted in a potentially similar setting to Ten Brink et al.'s study. The objective was to estimate the extent of the efficacy of 10-session PAT vs. 10-session Sham treatment, delivered over a two-week period, in addition to the standard rehabilitation program, in improving spatial neglect symptoms among individuals with moderate-to-severe spatial neglect. We aimed to examine two hypotheses:

- Hypothesis 1: Prism adaptation treatment reduced visuospatial symptoms of spatial neglect among patients in an inpatient setting providing intensive rehabilitation care.

- Hypothesis 2: Prism adaptation treatment enhanced the recovery of spatial neglect.

## Materials and methods

### Study design

A double-masked, randomized, sham-controlled trial was conducted to evaluate the efficacy of 10 once-daily sessions of PAT. The double-masked design was achieved as 1) patients were not informed about their group membership, no information about the mechanisms of goggles was discussed, and no questions about the treatment condition were raised, and 2) outcome measures were assessed by examiners who were masked from patients' group membership." The same double-masked design was previously used in other similar studies [42, 44]. The study was conducted according to the principles of the Declaration of Helsinki and was approved by the institutional ethics committee. Written informed consent was obtained from all participants. The individual on S1 Fig in this manuscript has given written informed consent (as outlined in PLOS consent form) to publish these case details.

### Setting

The study was conducted in a highly intensive inpatient comprehensive brain injury rehabilitation program (abbreviated as the BIR Program) in the Rehabilitation Center Kladruby, the Czech Republic.

## Participants

Patients consecutively admitted to the BIR Program (June 2017–July 2019) were invited to the study. Individuals admitted to the BIR Program (a) were in between 18 and 75 years of age, (b) obtained acquired brain injury (TBI or stroke), (c) had the brain injury no longer than 1 month since the time being discharged from acute care, (d) were able to participate in at least 4 hours of high intensity rehabilitation on a daily basis, receiving therapies from at least 2 of the four following specialized disciplines: physiotherapy, occupational therapy, psychology; speech and language therapy and (e) had cooperating family member with the rehabilitation team with expectation to return home given the prognosis and availability of the caregivers. In addition to admission criteria, study participation criteria were (f) presence of moderate or severe spatial neglect as indicated by the Catherine Bergego Scale (CBS > 10) via the Kessler Foundation Neglect Assessment Process (KF-NAP®) [46, 47] at baseline assessment, (g) confirmed unilateral brain injury, (h) physically and cognitively able to participate in PAT, and (i) able to provide informed consent. Participants were randomly allocated to the treatment or control group by the study coordinator using sealed envelopes with equal numbers of printed group assignment cards inside. The treatment group received PAT, and the control group received Sham treatment.

## Clinical information

With participants' informed consent, certain clinical information was extracted from the BIR Program records for the purpose of the present study. This included demographic information, results of standard measures upon admission indicating functional status in the motor and cognitive domains (such as the Modified Motor Subscale of the Functional Independence Measure (mFIM), Mini Mental State Examination (MMSE), and Berg Balance Scale (BBS)), and information about brain injury characteristics including injury description and brain lesion location. The lesion location was identified by an independent neurologist based on acute care medical records transferred to the BIR Program.

## Treatment

PAT was delivered, by a physiotherapist, using the treatment protocol and equipment of the Kessler Foundation Prism Adaptation Treatment (KF-PAT®) [48]. The procedures and equipment were the same for PAT and Sham treatment, except that the treatment group wore goggles fitted with 20-diopter prism lenses that shift the visual field to the ipsilesional side of space for 11.4 degrees of visual angle while the control group used flat goggles that did not shift the visual field at all. During each session, lasting approximately 15–20 minutes, participants completed 60 visuomotor movements while the first part of arm movements was blocked from view. The visuomotor movement was initiated from participants' chest toward a visual target (a 24.1-cm horizontal line or a 1-cm-diameter circle) printed at the center of a 29.7 x 21 cm paper sheet. Stimuli were pseudo-randomly presented either at body midline or in left or right space (32.1 cm to the side of body midline). The task was to mark the center of a line or cross out a circle. Before and after prism adaptation, pointing tasks were administered using the procedure described in previous studies [49]. If a participant did not demonstrate any after-effect for three consecutive PAT sessions, suggesting impaired cortico-cerebellar circuits, then they would be excluded from the study. If a participant reported discomfort during the session, the Nausea Profile [50] was administered to help evaluate potential adverse effects. Participants completed the 10-session, once-daily, treatment course over two weeks (skipping weekends), during morning hours, in addition to the therapy sessions provided in the BIR

Program. The treatment was always delivered as per the protocol, based on detailed documentations for fidelity check.

The BIR Program was a multidisciplinary, high-intensity, inpatient rehabilitation program offered up to 12 weeks to people with acquired brain injury. Patients' rehabilitation needs were assessed by the multidisciplinary rehabilitation care team during the first week of admission when an individualized rehabilitation plan and tailored goals were established using goal attainment scaling [51]. The goals were re-evaluated and adjusted every 3 weeks to meet changes in patients' needs and progress, and this occurred in multidisciplinary meetings. Patients typically participated in 6–10 rehabilitation sessions per day (a total of 4–5 hours), 6 days a week with less intensive program on Saturdays. The BIR Program included 30-minute daily computer-assisted cognitive training, weekly psychotherapy, 30-minute twice-daily sessions of functional independence training, 30-minute twice-daily sessions of physiotherapy (limb activation, balance training, walking training etc.), individual and group physical exercises in the gym (such as stretching, playing ball games, using stationary bikes etc.) or in the swimming pool (strength training, swimming etc.), 30-minute twice-daily speech and language therapy if needed, and various forms of recreational therapy (arts, crafts, etc.) If necessary, as deemed by the rehabilitation care team, patients are trained in further independence and compensatory strategies by nurse staff outside therapy hours. Patients could go home accompanied by their carers for several weekends, during which assessment of functional needs in home environment were conducted.

## Outcome measures

Outcome measures were administered by trained occupational therapists who were masked of participants' group membership, while treatment was delivered by different therapists. The assessments were performed at baseline (T1), after treatment (T2), two weeks after treatment (T3), and four weeks after treatment (T4). All time points were within the duration of participants' admission in the BIR Program. The assessments were carried out at the same location, all in one session, at the same time of each day, mostly by the same trained therapist (with exceptions when unavailable due to vacation etc., then another trained therapist would substitute).

**Catherine Bergego Scale (CBS) via Kessler Foundation Neglect Assessment Process (KF-NAP®) [46, 47].** The assessment consists of 10 categories: limb awareness, personal belongings, dressing, grooming, gaze orientation, auditory attention, navigation, collisions, having a meal, and cleaning after a meal. Each category is scored from 0 (no neglect) to 3 (severe neglect). The final score is calculated with the formula: (sum score ÷ number of scored categories) × 10 = final score [52]. The final score ranges from 0 to 30, and a positive score indicates the presence of spatial neglect.

**Bells test [53].** The Bells Test was printed on a sheet of 29.7 x 21 cm paper with 35 targets (bell-shaped figures) and other 280 distractors equally distributed in 7 columns. The paper was placed at the participant's midline. The participant was asked to circle the bells. The starting column and the number of circled bells were recorded. To ensure the coding consistency regardless of the side of neglect, the first column closest to the paper edge on the neglected side was coded as 1, and the coded value increased in the columns toward the non-neglect side of space. Spatial neglect is indicated when the discrepancy between the left and right omissions (excluding performance in the 4th column) is greater than 2, or when the coded value of the starting column is greater than 5 [54].

**Line bisection [55].** Participants were presented with five 20 cm horizontal line, each printed separately on a 29.7 x 21 cm paper sheet. One line was presented at one time right in

front of the participant, who was asked to mark the center of the line. The average deviation from the true center of the line was recorded. Deviation to the non-neglected side of space was coded positive and the neglect-sided negative. The criterion for spatial neglect is the deviation greater than 6.5 mm [54].

**Scene copying test [56].** Modified from a five-object figure copying test [6], the Scene Copying Test is a figure copying task consisting of two plants, a house, and two trees arranged in that order left-to-right, printed on the upper half of a page [56]. The task is to copy the entire "scene" to the lower half of the page. Each object is scored 2 (symmetric copy), 1.5 (partial omission of one side), 1 (complete omission of one side), .5 (complete omission of one side and partial omission of the other), or 0 (complete omission of the object or unrecognizable copy). The total score lower than 10 is considered abnormal [28]. In addition, each object is divided in half, resulting in 10 halves. For egocentric asymmetry, the two plants and the half of the house figure presented on the left side of the page are scored -1 for each half if copied, and the others on the right side of the page are scored +1 for each half if drawn. A partial omission results in .5 being deducted toward zero. For allocentric asymmetry, within each object (plant, house, or tree), the left half is scored -1 and the right +1 if it copied. The sum of an asymmetry greater than 0 indicates left-sided neglect in the given reference frame, and lower than 0 indicates right-side neglect.

## Analysis methods

All the analyses were performed using STATA/SE 16.1. All the significance level, alpha, was set to 0.05. We described participant characteristics using medians and interquartile ranges (IQRs) for continuous variables and counts for categorical variables. Group comparison of each variable was performed using the *U* test for continuous variables, and the chi-squared test for categorical variables. If any of the participant characteristics showed a significant difference between groups, they would be included to the main analyses that examined *a priori* hypotheses.

To examine Hypothesis 1 that PAT reduced visuospatial symptoms of spatial neglect, on each outcome measure, we included participants who showed spatial neglect in the measure at baseline (T1) and chose an analysis appropriate for the data distribution. For data of an outcome measure that was or could be transformed to be normally distributed (given its skewness and kurtosis), a mixed linear modeling (MLM) analysis was performed. MLMs offer higher power because they more accurately account for the within-subject variability by including the random intercepts and slopes [57]. While modeling subjects' intercepts and slopes as random effects, we included the independent variables of treatment group (PAT, Sham), assessment time (T1, T2, T3, T4), and the interaction of them as fixed effects while controlling for baseline performance (i.e., outcome measured at T1). For data of an outcome measure that could not be transformed into a normal distribution, we would inspect the pattern of the results and conduct appropriate analyses.

To examine Hypothesis 2 that PAT enhanced the recovery of spatial neglect, we inspected the occurrence rates of spatial neglect as defined by different measures in each time point among participants who showed a neglect symptom in a given test at T1. Thus, the spatial neglect occurrence rate at T1 was locked at 100% for each test, controlling for the sensitivity difference across tests. A lower occurrence rate after T1 indicated fewer patients with detected symptoms by a given measure, which would indicate a higher recovery rate. Chi-squared tests were conducted to examine the group differences.

## Results

### Participant characteristics

A total of 304 potential participants were screened for eligibility (Fig 1). Thirty-four (11.2%) participants meeting inclusion criteria were randomized into the PAT group or the Sham

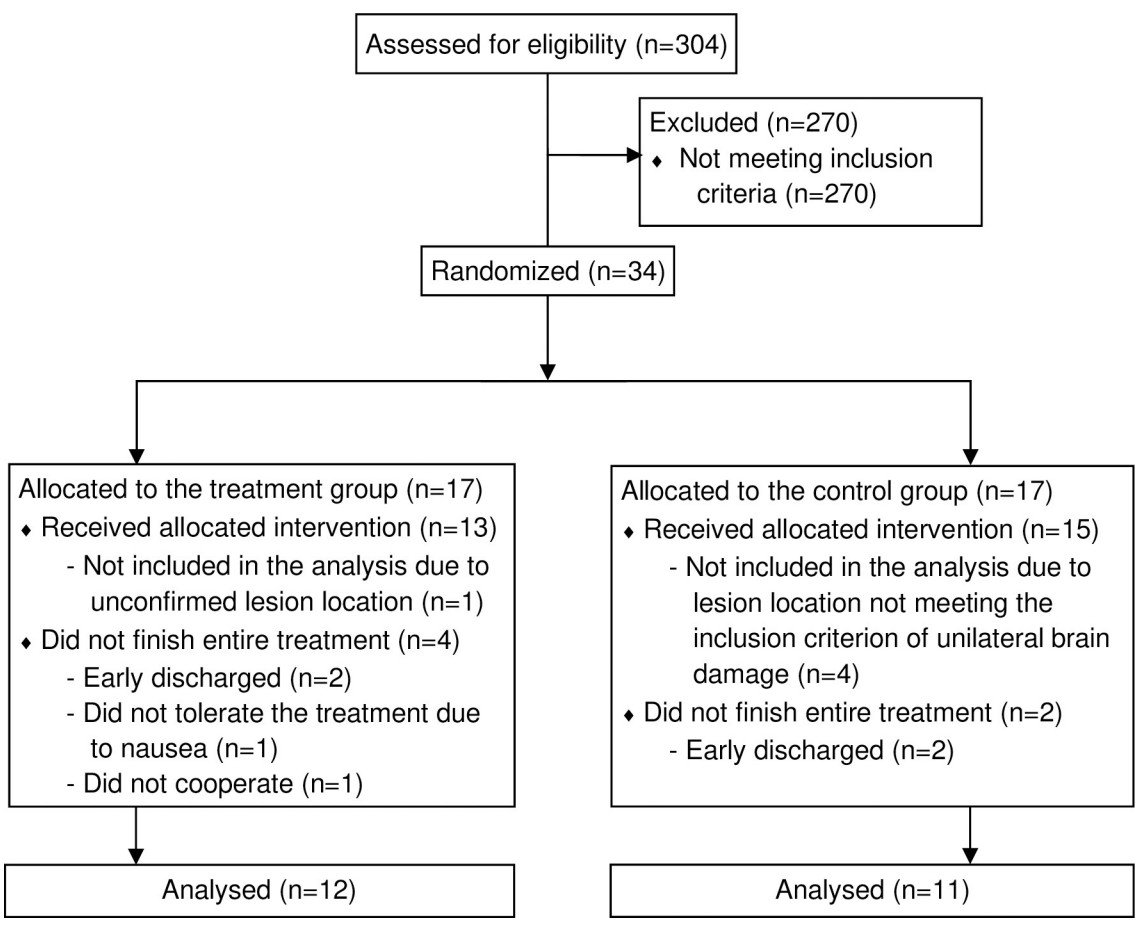

**Fig 1. Flow chart of the randomized feasibility and pilot trial.**

group (see S1 Table). Twenty-three (67.6%) participants completed all the treatment and assessment sessions and were included in the analysis As summarized in Table 1, there was no significant difference in any of the participant demographic characteristics, functional status at admission, or severity level of spatial neglect at baseline as measured using the CBS via KF-NAP.

Table 2 presents lesion locations in each participant. 13 (56.5%) participants had lesions in both cortical and subcortical regions, 7 (30.4%) had lesions in the cortical areas sparing the subcortical structures, and 3 (13.0%) sustained lesions confined in the subcortical areas. Table 2 also includes a summary of the presentation of spatial neglect based on different outcome measures' criteria. While all the participants were enrolled to the study for their presentation of moderate to severe neglect determined using the CBS via KF-NAP, 5 (21.7%) participants demonstrated neglect in all the outcome measures at baseline and 1 (4.3%) showed no neglect in any of the measures (Table 2).

### Hypothesis 1: Prism Adaptation Treatment (PAT) reduced visuospatial symptoms of spatial neglect

**Catherine Bergego Scale (CBS) via KF-NAP.** All the participants (12 in the PAT group and 11 in the Sham group) were included in this analysis as they were enrolled to the study based on the CBS score > 10. A mixed linear modeling (MLM) analysis was conducted to

**Table 1. Participant characteristics presented in counts for categorical variables and in medians (IQRs) for continuous variables.**

| Variable | PAT group (n = 12) | Sham group (n = 11) | Group comparison (p value) |
|---|---|---|---|
| **Sex** (male/female) | 5 / 7 | 5 / 6 | .855 |
| **Age** (in years) | 51.5 (47.5–55) | 58 (53–61) | .102 |
| **Formal education** (in years) | 12.5 (12–13) | 12 (11–13) | .237 |
| **Injury type** (stroke / TBI) | 11 / 1 | 10 / 1 | .949 |
| **Neglected side** (left/right) | 11 / 1 | 9 / 2 | .484 |
| **Time post injury/stroke at admission** (in days) | 58 (38.5–74) | 48 (35–79) | .735 |
| **Time post injury/stroke at the first PAT session** (in days) | 76 (69–133.5) | 70 (62–97) | .424 |
| **mFIM at admission** (range 7–47; higher = better function) | 23 (18–24.5) | 20 (19–25) | .853 |
| **MMSE at admission**[*] (range 0–30; higher = better function) | 19 (18–27); n = 10 | 19 (17–21.5); n = 8 | .691 |
| **BBS at admission** (range 0–56; higher = better function) | 5 (4–15) | 5 (4–25) | .827 |
| **CBS via KF-NAP at baseline** (range 0–30; higher = greater impairment) | 13 (12. 5–20) | 14 (12–17) | .781 |

Abbreviations: PAT, prism adaptation treatment; mFIM, Modified Motor Subscale of the Functional Independence Measure; MMSE, Mini Mental State Examination; BBS, Berg Balance Scale; CBS, Catherine Bergego Scale; KF-NAP, Kessler Foundation Neglect Assessment Process.

[*]MMSE was unavailable for five participants, and the numbers of participants were noted.

examine the square-root-transformed CBS as described in the Methods section. Assessment time was coded as a continuous variable. The only significant effect was assessment time, b = -0.50, SE = 0.10, p < .001, 95%CI = [-0.70, -0.31], while treatment group, b = -0.25, SE = 0.35, p = 0.473, 95%CI = [-0.93, 0.43], or the treatment x time interaction, b = 0.09, SE = 0.15, p = 0.539, 95%CI = [-0.21, 0.40], did not show significant effects. To examine whether there were significant changes between assessment sessions, we performed MLM again but with assessment session as a categorical variable (T1, T2, T3, T4). The results showed significant effects of all assessment sessions (reference: T1), all p values < 0.001, but there was no significant effect of treatment group, p = 0.947, or interaction effect, all p values > 0.3. Thus, both groups improved from T1 to T4 to the similar extent, and there was no specific effect of PAT (Fig 2A).

**Bells test.** Twenty participants (11 in the PAT group and 9 in the Sham group) were included in this analysis as they showed neglect in one of the measures performed on the Bells Test at baseline. Based on the lateralized omission discrepancy criterion, 19 participants (11 in the treatment group and 8 in the control group) omitted more than 2 targets on the contralesional side than on the ipsilesional side of space. Because the data could not be transformed into a normal distribution, the planned MLM was not performed. Instead, we calculated the linear regression coefficient between lateralized omission discrepancy and assessment session within each participant to indicate the performance trajectory over time, in which an improvement was noted with a negative regression. Given that this generated dataset was normally distributed, we contacted a *t* test and found no significant difference, effect size d = 0.35, p = 0.481, between the mean regression, -1.92 (SD = 1.08), in the PAT group and that, -1.55 (SD = 1.14), in the Sham group. Nonetheless, the PAT group's trajectory was significantly lower than zero, p < 0.001, suggesting improvement, so was the Sham group's, p = 0.006. While inspecting the plotted results (Fig 2B), we observed the most apparent difference was at T2, thus a *U* test was conducted to compare the difference between the groups, which was found not reaching the significance level, z = 1.08, p = 0.280, effect size r = 0.25.

Based on the starting column criterion for spatial neglect, 20 participants (11 in the treatment group and 9 in the control group) were included. The median value in any condition

**Table 2. Participant brain lesion locations and presentation of spatial neglect at baseline, based on different test criteria.**

| Group | ID* | Injured hemisphere | Cortical | | | | | Subcortical | | Bells Test | | Line Bisection | Scene Copying | |
|---|---|---|---|---|---|---|---|---|---|---|---|---|---|---|
| | | | F | T | P | O | In | BG | Th | L-R | Start | | Ego | Allo |
| PAT | 1 | R | | x | x | | | x | | x | | | | |
| PAT | 2 | R | x | x | x | | | | | x | x | x | x | x |
| PAT | 3 | R | | x | x | | | x | | x | x | | | |
| PAT | 4 | R | | | | | | x | | x | x | x | | |
| PAT | 5 | R | x | x | x | | | | | x | x | x | | |
| PAT | 6 | R | x | x | | | | x | | x | x | | x | x |
| PAT | 7 | R | x | x | x | | | x | x | x | x | | x | x |
| PAT | 8 | R | x | | x | | | | | x | x | | | |
| PAT | 9 | R | x | x | x | x | x | x | x | x | x | x | x | |
| PAT | 10 | R | x | x | x | | | x | | x | x | x | x | x |
| PAT | 11 | R | | | | | | x | | x | x | | | |
| PAT | 12 | L | | x | x | x | | x | | | x | | | |
| Sham | 13 | R | x | x | x | x | x | x | x | x | x | x | x | |
| Sham | 14 | R | x | x | x | x | x | x | x | x | x | x | x | x |
| Sham | 15 | R | | x | x | | x | x | | | x | x | | |
| Sham | 16 | R | x | | x | x | | | | x | x | x | x | x |
| Sham | 17 | R | x | | | | | | | x | x | | x | x |
| Sham | 18 | R | x | x | x | | | | | x | x | | | |
| Sham | 19 | R | | | | | | x | | | x | | | |
| Sham | 20 | R | | x | x | x | | | | x | x | x | x | x |
| Sham | 21 | R | x | x | x | | | x | | x | | x | x | x |
| Sham | 22 | L | x | x | x | | | x | x | x | x | | x | x |
| Sham | 23 | L | | x | x | | | x | | | | | | |

Abbreviations under the Cortical header: F, frontal; T, temporal; P, parietal; O, occipital; In, insular. Abbreviations under the Subcortical header: BG, basal ganglia; Th, thalamus; Abbreviations for tests: L-R, lateralized discrepancy; Start, starting column; Ego, egocentric; Allo, allocentric.

*ID numbers were generated for the purpose of data presentation and did not reflect the chronical order of study participation or personal identity.

from the combination of treatment group and assessment session was 7 (the column far to the neglected side of space) with varied IQRs. This suggested a highly skewed dataset and required a larger sample size for a statistical analysis. Overall, both groups improved in lateralized bias from T1 to T4, and there were no consistent PAT effects on the Bells Test performance.

**Line bisection.** Eleven participants (5 in the PAT group and 6 in the Sham group) were included. An MLM was conducted to examining the log-transformed line bisection deviation as described in the Methods section. No effect was found significant, all p values > 0.1, when assessment time was entered to the model as a continuous variable. To examine whether there was an effect between assessment sessions, we conducted a separate MLM but with assessment session as a categorical variable (T1, T2, T3, T4) with T1 as the reference. The results showed a significant effect of T3, p = 0.05, but there was no significant effect of other time point, treatment group, or interaction effect, all p values > 0.1. Thus, both groups showed potential changes from T1 to T3, and there was no effect of PAT (Fig 2C).

**Scene copying.** In the twelve participants (5 in the PAT group and 7 in the Sham group) who showed egocentric and/or allocentric neglect on this measure at baseline, we conducted an MLM on the total accuracy score using the same structure as described above. Similar results were found such that assessment time was the only variable that showed a significant effect: As a continuous or categorical variable, the effect of assessment time was significant, all

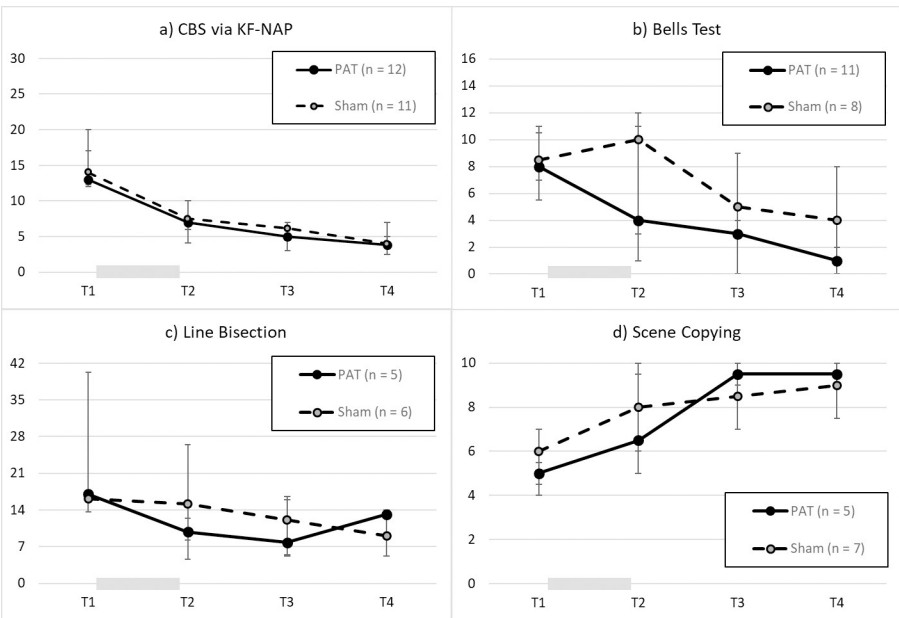

**Fig 2.** Results, from T1 to T4, each two weeks apart, of a) CBS via KF-NAP, b) lateralized omission discrepancy in the Bells Test, c) deviation (mm) in line bisection, and d) accuracy score of the Scene Copying test. The gray bar between T1 and T2 indicates the treatment period. Each data point denotes the median with the error bar representing the IQR.

p values < 0.006, while the other variables were not, p > 0.1. Thus, both groups improved in copying the figures from T1 to T4 with no effect of PAT (Fig 2D).

### Hypothesis 2: Prism Adaptation Treatment (PAT) enhanced the recovery of spatial neglect

One participant of the PAT group (n = 12) scored 0 on the CBS at T2 and at T3, and two scored 0 at T4. However, none of the Sham group (n = 11) scored 0 at any time point. None-theless, none of the group comparisons reached significance, all p values > 0.1. The same set of analyses were performed based on the criteria of the lateralized discrepancy omission and starting column of the Bells Test, deviation of Line Bisection, and egocentric and allocentric asymmetries of Scene Copying Test. The results are presented in Table 3. None of the analyses yielded a significant result although at each time point the occurrence rate of spatial neglect in

**Table 3. Chi-squared results on the occurrence rates of spatial neglect in the treatment vs. control groups, based on the criteria set in different measures.**

| Assessment time | CBS via KF-NAP | Bells Test | | Line Bisection | Scene Copying | |
| --- | --- | --- | --- | --- | --- | --- |
| | | L-R | Start | | Ego | Allo |
| T2 | $X^2 = 0.96$ | $X^2 = 0.01$ | $X^2 = 0.13$ | $X^2 = 0.75$ | $X^2 = 0.75$ | $X^2 = 0$ |
| | $p = 0.328$ | $p = 0.912$ | $p = 0.714$ | $p = 0.387$ | $p = 0.387$ | $p = 1$ |
| T3 | $X^2 = 0.95$ | $X^2 = 2.30$ | $X^2 = 0.90$ | $X^2 = 0.05$ | $X^2 = 0.05$ | $X^2 = 0.09$ |
| | $p = 0.329$ | $p = 0.129$ | $p = 0.343$ | $p = 0.819$ | $p = 0.819$ | $p = 0.764$ |
| T4 | $X^2 = 1.83$ | $X^2 = 1.17$ | $X^2 = 0.04$ | $X^2 = 0.75$ | $X^2 = 0.40$ | $X^2 = 0.09$ |
| | $p = 0.176$ | $p = 0.280$ | $p = 0.845$ | $p = 0.387$ | $p = 0.527$ | $p = 0.764$ |

Abbreviations: CBS, Catherine Bergego Scale; KF-NAP, Kessler Foundation Neglect Assessment Process; L-R, lateralized discrepancy; Start, starting column; Ego, egocentric; Allo, allocentric.

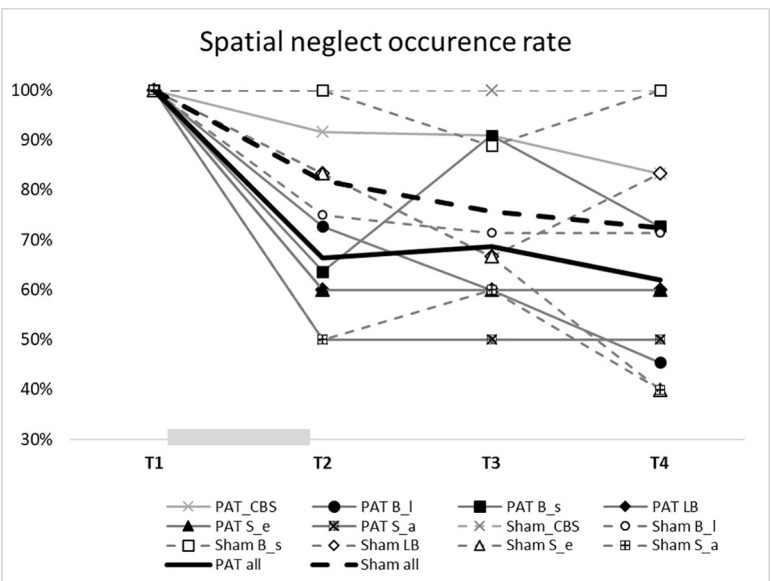

**Fig 3. Occurrence rates of spatial neglect at each time point based on the criterion of a given outcome measure.**
The gray bar between T1 and T2 indicates the treatment period. CBS: Catherine Bergego Scale; B_l: Bells Test–lateralized omission discrepancy; B_s: Bells Test: starting column; LB: Line Bisection; S_e: Scene Copying–egocentric asymmetry; S_a: Scene Copying–allocentric asymmetry.

the PAT group was lower than the Sham group in all the measures except for the allocentric asymmetry in the Scene Copy Test. On average, 39.1% of the PAT group and 28.6% of the Sham group recovered from spatial neglect (Fig 3).

## Discussion

In the present study, we conducted an RCT in an inpatient setting where patients received a highly intensive comprehensive brain injury rehabilitation program. Both the PAT and Sham groups improved in visuospatial ability (assessed using paper-based tests) as well as visuospatial function (assessed using an ecological assessment) to the similar extents that could not be differentiated statistically. In addition, the PAT group appeared recovering better than the Sham group, but the comparison was under powered. Thus, we found no evidence that PAT specifically reduced spatial neglect symptoms or enhanced spatial neglect recovery.

The present study replicated Ten Brink et al. [44] using the same PAT treatment regimen (10 once-daily sessions over two weeks) in a similar setting where patients received intensive neurorehabilitation that had specific emphasis on spatial neglect. Unlike Ten Brink et al. who included patients regardless their neglect severity, we recruited patients with moderate to severe neglect, which would prevent ceiling effects when evaluating improvement. Nonetheless, our findings were comparable to theirs. This suggests that visuomotor training with sham/flat goggles was effective to certain extent, and that intensive rehabilitation programs that had been in place in both studies may have been successful in improving visuospatial abilities and functional outcomes related to spatial neglect.

Several researchers have noted that sham treatment may have beneficial effects on reducing neglect symptoms. Serino and colleagues [58] are among the first who offered an account for the mechanisms underlying the improvement after sham treatment. The visuomotor exercise during PAT is a task that requires motor planning and execution guided by a visual target, i.e., initiating and performing arm reaching movements toward a visible target, which relies on

eye-hand coordination. Patients perform this task repeatedly during a session (60 times in our study protocol and 90 times in Serino et al.), and sometimes they reach toward the neglected side of space. Serino et al. [58] postulated that this orienting behavior toward the neglected side could be reinforced by repetition within a session and over multiple sessions, leading to amelioration of spatial neglect. In addition, sustained attention may be strengthened during the repetitive task guided by the therapist. Even though the task seems easy and effortless, patients are mentally present not only at the moment when they perform an arm movement to a visual target but also during a sustained period of time, ideally the entire session. This mental state of being present may strengthen the ability to sustain attention, which in turn, facilitate improvement of other attention-related abilities such as spatial attention and thus ameliorate spatial neglect [59–61]. Further research is required to understand the "active ingredients" of PAT and sham treatment.

Several possibilities may explain why the current study setting, i.e., the BIR Program, was effective for ameliorating spatial neglect symptoms. The most likely possibility is that the BIR Program provided a variety of evidence-based therapeutic elements for spatial neglect, applied throughout the entire rehabilitation stay. This included elements of visual scanning training, optokinetic stimulation, limb activation, constraint induced movement therapy (for a review of these methods, see [62–64]). Visual scanning training was highly emphasized during therapy sessions for mobility and activities of daily living (ADL) trainings. Moreover, patients in the BIR Program received 30-minute daily computer-assisted cognitive training on RehaCom (Hasomed, Magdeburg, Germany) and CogniPlus (Schuhfried, Moedling, Austria) platforms. These cognitive training sessions were designed to target the core cognitive deficits in domains such as spatial attention and spatial working memory through computerized optokinetic stimulation [62] or training for spatial working memory (for discussion of association between spatial neglect and spatial working memory, see [65]). In addition, twice daily physiotherapy sessions in the BIR Program usually involved limb activation (with the help of physiotherapist and/or robotic systems) which was found to reduce spatial neglect even when performed passively [66]. Some patients in the BIR Program received other forms of physiotherapeutic interventions such as constraint induced movement therapy or mirror therapy, which was found effective reducing spatial neglect severity [67, 68]. Because a combination of different interventions was more effective than a single approach in treating spatial neglect [69], any benefit from PAT may have been masked by the BIR Program.

In addition to treatment outcomes, the present study also demonstrated the difficulty in conducting an RCT in an inpatient rehabilitation setting. Only one in ten patients were eligible for the study, and one third of study participants could not complete the study. This suggests a suboptimal feasibility in terms of research participant recruitment and of implementing a daily research treatment protocol on top of an already busy schedule of therapies. We recommend that future large-scale definitive RCTs collaborate with multiple centers to increase the chance of reaching a necessary sample size and coordinate with the therapy team to incorporate PAT and Sham treatment within scheduled therapy sessions.

## Study limitations

The major limitations of our study are the small sample size and the heterogeneity of the sample. Participants in our study had different acquired brain injury etiology (stroke or TBI), neglected side of space (left or right), and lesion locations. PAT is not equally effective for all patients. For example, several studies have found lesions involving the frontal lobe [26, 70–72], the temporal lobe [73, 74], the parietal lobe [26, 72–74], or the occipital lobe [31] led to different PAT responses. Our sample size was too small to examine lesion-based hypotheses.

However, the RCT design should have eliminated some data noise from the heterogeneous sample. Nonetheless, the sample size was not large enough to enable us to perform parametric analyses on several outcome measures, limiting result interpretation and further hypothesis generation.

## Conclusion

The present findings suggest that both PAT and Sham treatment may improve visuospatial ability and function among individuals with spatial neglect after unilateral brain damage. We found PAT not particularly effective in ameliorating spatial neglect in the clinical setting that offers highly intensive rehabilitation program that has emphasized evidence-based practice addressing spatial neglect. Further investigations are needed to understand the mechanisms underlying PAT and Sham treatment in order to refine the treatment or create new interventions that will target specific attention, visuospatial, or motor control functions critical to spatial neglect.

## Supporting information

**S1 Fig. PAT setup.**
(TIF)

**S1 Table. Randomized participant characteristics.**
(TIF)

**S1 Dataset. Raw data underlying the plots in Figs 2 and 3.**
(XLSX)

## Author Contributions

**Conceptualization:** Tomas Vilimovsky, Peii Chen, Pavel Harsa.

**Data curation:** Tomas Vilimovsky, Peii Chen, Kristyna Hoidekrova.

**Formal analysis:** Peii Chen.

**Investigation:** Kristyna Hoidekrova, Jakub Petioky.

**Methodology:** Tomas Vilimovsky, Peii Chen.

**Project administration:** Tomas Vilimovsky, Kristyna Hoidekrova, Jakub Petioky.

**Resources:** Tomas Vilimovsky, Kristyna Hoidekrova, Jakub Petioky, Pavel Harsa.

**Supervision:** Pavel Harsa.

**Visualization:** Peii Chen.

**Writing – original draft:** Tomas Vilimovsky, Peii Chen.

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
