## [Decision Letter · Decision Letter 0]

13 Nov 2020

PONE-D-20-30567

Prism adaptation treatment to address spatial neglect in an intensive rehabilitation program: A randomized controlled trial

PLOS ONE

Dear Dr. Vilimovský,

Thank you for submitting your manuscript to PLOS ONE. We have now received three expert reviews and all three agreed that it was a well-conducted study that adds to the knowledge on neglect rehabilitation. Each of the reviewers had minor suggestions for clarification and improvement. Therefore, we invite you to submit a revised version of the manuscript that addresses the points raised during the review process.

We look forward to receiving your revised manuscript.

Kind regards,

Daniel Mirman

Academic Editor

PLOS ONE

2.) Please ensure that you include a title page within your main document. We do appreciate that you have a title page document uploaded as a separate file, however, as per our author guidelines (http://journals.plos.org/plosone/s/submission-guidelines#loc-title-page) we do require this to be part of the manuscript file itself and not uploaded separately.

3.) We note that you have a patent relating to material pertinent to this article. Please provide an amended statement of Competing Interests to declare this patent (with details including name and number), along with any other relevant declarations relating to employment, consultancy, patents, products in development or modified products etc. Please confirm that this does not alter your adherence to all PLOS ONE policies on sharing data and materials, as detailed online in our guide for authors http://journals.plos.org/plosone/s/competing-interests by including the following statement: "This does not alter our adherence to  PLOS ONE policies on sharing data and materials.” If there are restrictions on sharing of data and/or materials, please state these. Please note that we cannot proceed with consideration of your article until this information has been declared.

Reviewers' comments:

Reviewer's Responses to Questions

**Comments to the Author**

1. Is the manuscript technically sound, and do the data support the conclusions?

Reviewer #1: Partly

Reviewer #2: Yes

Reviewer #3: Yes

2. Has the statistical analysis been performed appropriately and rigorously? 

Reviewer #1: Yes

Reviewer #2: Yes

Reviewer #3: Yes

3. Have the authors made all data underlying the findings in their manuscript fully available?

Reviewer #1: Yes

Reviewer #2: No

Reviewer #3: Yes

4. Is the manuscript presented in an intelligible fashion and written in standard English?

Reviewer #1: Yes

Reviewer #2: Yes

Reviewer #3: Yes

5. Review Comments to the Author

Reviewer #1: The authors conducted an RCT to compare prism adaptation with sham adaptation as a treatment for neglect. Prism adaptation is the most widely studied neglect treatment, although results have been mixed. The first studies on this matter included small sample sizes, did not include functional outcome measures, and/or did not include follow up measurements. In the current study, 23 neglect patients were included and randomly assigned to either prism or sham adaptation. Outcome measures included a functional instrument and paper-and-pencil tasks, all commonly used tests for neglect. Patients were tested up to 4 weeks after treatment. The authors used mixed linear modeling to analyse their results, which is a strength of the study. Results showed no significant difference between the treatment groups. The study has been set-up thoroughly and results add to the knowledge on neglect rehabilitation. I only have some minor suggestions.

• The authors refer to the sham treatment as “Sham PAT”. However, as PAT stands for “Prism adaptation treatment”; adding “sham” is technically incorrect as there are no prisms in the sham treatment. It would therefore be better to refer to “Prism adaptation treatment” and “Sham treatment” or “Sham adaptation treatment”

• The authors included patients with right lesions and left lesions. This is not so common in previous research on this topic, maybe the authors could already mention this in the introduction section.

• The authors nicely specify their hypotheses. However, the difference between them is not clear to me, what is the difference between “reduced visuospatial symptoms of spatial neglect” versus “enhanced the recovery of spatial neglect”?

• Page 3, line 60: here might be a type, “to” instead of “two”.

• P4 line 92: “double-masked” is a debated concept in prism adaptation trials, because the person who has to put on the goggles can see by the shape of the goggles which condition it is. Also, patients could observe the visual shift (even though they don’t know what it means). Maybe only the effect evaluator was different from the person who provided the treatment? Please specify.

• Page 5 line 98-99 contains a typo: patients were in between 18 and 75 years of age

• The authors describe the rehabilitation program very well. Could the explicitly mention whether specific neglect treatment was provided or not?

• In the analyses section at page 9, the authors very nicely link their analyses to their hypotheses. Could the authors maybe repeat the hypotheses here?

• In the flowchart it says that one patient did not finish the entire treatment because he or she “did not tolerate treatment”. Was this treatment-specific (e.g. nausea because of the goggles) or because of general reasons (e.g. fatigue or motivation)?

• In Table 1, could the authors provide all statistics instead of only the p-values (also effect sizes if possible).

• In the figure caption of Figure 2, could the authors remind the readers when the treatment was provided (between which measure moments); and how much time was in between the measure moments?

• In the discussion section on page 16, the authors write that the training with sham/flat goggles were effective and that the intensive rehabilitation program was successful. However, the present study does not support this statement, as all of it could have been due to spontaneous recovery. Of course it can be hypothesized that those elements were effective, but it has not been proven here. Therefore, the statements should be toned down and the possibility of spontaneous recovery should be mentioned. This also holds for the conclusion.

• In the discussion section I miss some specific suggestions or considerations for future research. For example, studies have tried to disentangle which patients (e.g. without specific lesions) would benefit from prism adaptation. Do the authors think that is the best way to continue this line of research? Should prism adaptation be compared with a real placebo treatment that is not expected to be effected? Etc.

Reviewer #2: Abstract:

It would be helpful to include whether participants had neglect due to stroke or any brain injury.

Introduction:

Line 42: There is evidence to suggest the after effect of prism adaptation lasts for hours (e.g. Turton 2010)

Methods:

Design: please state what phase of study this was, i.e. phase II or III, pilot study etc

Line 98: Please state what BIR stands for clearly, whether this is an inpatient setting, and the country of the study

Participants: it would be clearer if you combined the inclusion criteria for the BIR programme with the inclusion criteria for the study

At what time point were participants assessed as meeting the inclusion criteria with the KF-NAP, was this in addition to the baseline assessment or is this done on admission?

Line 113: please clarify that the outcome assessors were different people to those who delivered the intervention

Line 130: please state how long each PAT session lasted.

Treatment: please state who delivered the treatment and whether it was always delivered as per the protocol.

Outcome measures: please confirm whether the KF-NAP was carried out as per the assessment instructions i.e. same time of day each time, same location, all in one session.

Analysis: please state whether analysis was completed on an intention to treat basis

Results:

The baseline data presented in table 2 is quite confusing, it is not clear what ‘start’ means or how it is relevant, or why ego/allocentric is reported for scene copying but not for the bells test. Actual summary test scores would be more useful.

It would be helpful to include all relevant statistics e.g. means, medians etc in a table as well as the graphs in order to aid interpretation for both hypothesis. This is required for journal policy.

Discussion:

Line 337: Ten Brink used 10 dioptre prisms but you used 20 dioptre, this needs acknowledging

Your point about all the different interventions patients received is interesting – the Cochrane review of spatial neglect covers all these types of treatment so it sounds like a bit of everything is best. Do you have a breakdown of types of other interventions patients received e.g. amount of limb activation etc?

You conclude that more study into PAT is needed – it would be interesting to test the neglect outcomes of patients in your programme generally to see if intensive combined treatment is the most effective treatment and perhaps research into PAT as a treatment for neglect alone needs to stop. Just a general thought based on your interesting findings.

Reviewer #3: Thank you so much for this study and the manuscript. Very interesting and great work, high quality RCT (somewhat limited by small sample size and power). Overall, it is very well written. I have a few minor suggestions and food for thought:

1) Abstract: please specify the comparison intervention (vs. sham PAT) and population (patients with moderate-severe post-stroke SN).

2) Introduction: Similar comment to above, please state a clear PICO objective. e.g. The objective was to estimate the extent of the efficacy of a PAT program (nature, frequency, duration) vs. sham PAT program (xx) in improving xx among individuals with mod-sev post-stroke SN.

3) P7-8 missing references for CBS and Scene Copying Test.

4) Treatment: Please restate frequency and duration clearly in this paragraph. Add a picture of treatment setup (maybe as an appendix) and the sheets that were used for visual targets (was it computerized?).

5) Results: please start by how many were recruited and randomized, referring to your flow chart and only then proceed to discuss those what were analyzed. Also Table 1 must include info on all initially recruited 34 patients who were randomized (not sure why randomizing if a patient is not meeting your criteria re lesion location) and not only those who were included in the final analysis.

6) Specify that your analysis was NOT intention-to-treat since you did not include those who did not finish the program in your analysis. It was per protocol.

7) Important information for clinicians looking into adopting this treatment is who is the supplier for the prisms that were used. Can you indicate please. Also, can the treatment setup be accessed (e.g. existing computer program or existing treatment sheets that can be accessed by clinicians).

I wonder if another factor that have led to similar results in both groups is that your patients were mostly in acute/subacute phase of stroke recovery - where the window for recovery is of course larger. This can be one of your justifications and also a rationale to replicate such a study with chronic stroke patients (where we need more work as SN is known to persist in chronic phases of stroke recovery).

Thank you, good luck, great work!

6. PLOS authors have the option to publish the peer review history of their article (what does this mean?). If published, this will include your full peer review and any attached files.

Reviewer #1: No

Reviewer #2: **Yes: **Verity Longley

Reviewer #3: No

---

## [Author Response · Author response to Decision Letter 0]

29 Dec 2020

Dear Dr. Mirman,

Thank you for the helpful suggestions from the three reviewers on our manuscript titled “Prism adaptation treatment to address spatial neglect in an intensive rehabilitation program: A randomized controlled trial” (Manuscript Number: PONE-D-20-30567). All reviewers helped indicate sections where we needed to write more clearly, or in more detail, to rectify several misunderstandings. We hope you agree we have greatly improved the paper in line with all the suggestions, itemized below. Changed texts are highlighted in yellow in the revised manuscript. 

Following amended statement of Competing Interests was added to cover letter: Following patent was used in this study: Wearable System and Methods of Treatment of a Neurocognitive Condition (U.S. Patent Number 10,739,618). Patent owner: Kessler Foundation. This does not alter our adherence to PLOS ONE policies on sharing data and materials.

Reviewer 1:

1. The authors refer to the sham treatment as “Sham PAT”. However, as PAT stands for “Prism adaptation treatment”; adding “sham” is technically incorrect as there are no prisms in the sham treatment. It would therefore be better to refer to “Prism adaptation treatment” and “Sham treatment” or “Sham adaptation treatment” 

Response: As suggested, we changed “Sham PAT” to “Sham treatment” or “Sham” depending on the context (for example, see line 30).

2. The authors included patients with right lesions and left lesions. This is not so common in previous research on this topic, maybe the authors could already mention this in the introduction section. 

Response: We clarified the rationale of participant selection in the Introduction section (see line 113).

3. The authors nicely specify their hypotheses. However, the difference between them is not clear to me, what is the difference between “reduced visuospatial symptoms of spatial neglect” versus “enhanced the recovery of spatial neglect”?

Response: Improvement can be defined several ways. The two hypotheses were testing improvement in two ways. Hypothesis 1 was focused on improvement measured as degrees of neglect severity (quantified using test scores). Hypothesis 2 was focused on improvement defined in dichotomy (recovered vs. not), in which recovery is indicated by no detectable symptom of spatial neglect on a given test.

4. Page 3, line 60: here might be a type, “to” instead of “two”.

Response: The meaning of “to” here refers to “compare something to something”.

5. P4 line 92: “double-masked” is a debated concept in prism adaptation trials, because the person who has to put on the goggles can see by the shape of the goggles which condition it is. Also, patients could observe the visual shift (even though they don’t know what it means). Maybe only the effect evaluator was different from the person who provided the treatment? Please specify.

Response: Thank you for this comment. We agree that “double-masked” concept might be debatable in this case. So we tried to make the information more clear and we have included this information to the Study Design section (see lines 125 - 129): “The double-masked design was achieved as 1) patients were not informed about their group membership, no information about the mechanisms of goggles was discussed, and no questions about the treatment condition were raised, and 2) outcome measures were assessed by examiners who were masked from patients’ group membership..” 

The same double-masked design was previously used in other similar studies (Ten Brink et al., Neurorehabil Neural Repair, 2017; Mizuno et al., Neurorehabil Neural Repair, 2011).

6. Page 5 line 98-99 contains a typo: patients were in between 18 and 75 years of age.

Response: Corrected

7. The authors describe the rehabilitation program very well. Could the explicitly mention whether specific neglect treatment was provided or not?

Response: Thank you for the suggestion. Because no two patients received the same therapy regime in the BIR program, we prefer not to highlight any specific neglect treatment in the Methods section. Thus, in the previous and current manuscripts, we included in the information in the Discussion (see lines 407 - 425).

8. In the analyses section at page 9, the authors very nicely link their analyses to their hypotheses. Could the authors maybe repeat the hypotheses here?

Response: Thank you for this suggestion. We repeated the hypotheses in this section (see lines 245 and 256).

9. In the flowchart it says that one patient did not finish the entire treatment because he or she “did not tolerate treatment”. Was this treatment-specific (e.g. nausea because of the goggles) or because of general reasons (e.g. fatigue or motivation)?

Response: This was because of nausea. We added this information to the flowchart.

10. In Table 1, could the authors provide all statistics instead of only the p-values (also effect sizes if possible).

Response: Table 1 presents patients’ characteristics by group. Their characteristics were facts independent of treatment assignment. The purpose of comparing each variable was to demonstrate any reliable difference between groups, rather than showing an “effect” due to treatment. Thus, presenting effect sizes is not appropriate. 

11. In the figure caption of Figure 2, could the authors remind the readers when the treatment was provided (between which measure moments); and how much time was in between the measure moments?

Response: A gray bar was added to each chart in Figures 2 and 3 to indicate the treatment period. We also revised the figure caption as suggested. 

12. In the discussion section on page 16, the authors write that the training with sham/flat goggles were effective and that the intensive rehabilitation program was successful. However, the present study does not support this statement, as all of it could have been due to spontaneous recovery. Of course it can be hypothesized that those elements were effective, but it has not been proven here. Therefore, the statements should be toned down and the possibility of spontaneous recovery should be mentioned. This also holds for the conclusion.

Response: Thank you for this comment. As pointed out, spontaneous recovery might play a role in the present study because all participants received therapies provided by the rehabilitation hospital and by the research team. To examine the role of spontaneous recovery, a third arm would have been needed in the trial, and in such group patients would receive no therapist contact or any intervention at all. The advantage of the current study design (randomized, sham-controlled trial) and analysis model (MLM) is that noises within the data related to any random variables (such as spontaneous recovery, cognitive aging, underlying learning ability, visuomotor function in upper limbs, therapy engagement, etc.) are equally accounted in both PAT and Sham groups. Therefore, from the perspective of methodology, we do not think spontaneous recovery played a role significantly enough to be highlighted in the discussion. 

13. In the discussion section I miss some specific suggestions or considerations for future research. For example, studies have tried to disentangle which patients (e.g. without specific lesions) would benefit from prism adaptation. Do the authors think that is the best way to continue this line of research? Should prism adaptation be compared with a real placebo treatment that is not expected to be effected? Etc.

Response: Given the small sample size of the current study, the study is considered a randomized feasibility and pilot trial. Following the CONSORT guidelines (Eldridge et al., 2016), we added a paragraph of recommendations to future definitive large-scale RCTs. However, it might be unethical to recommend a no-treatment condition in research because PAT is recommended in several clinical practice guidelines. Also, what can be a “real placebo” condition for PAT? Placebo means that patients in this condition receive, by appearance, the same treatment but without active ingredients of the treatment. By this definition, Sham treatment provided in the present study is placebo, isn’t it? We are confident that the design and conduct of the present study was rigorous and systematic examining a priori hypotheses and informing future definitive RCT design. 

Reviewer #2: 

14. Abstract:

It would be helpful to include whether participants had neglect due to stroke or any brain injury.

Response: We included this information as suggested (see line 33).

15. Introduction:

Line 42: There is evidence to suggest the after effect of prism adaptation lasts for hours (e.g. Turton 2010)

Response: We corrected the sentence as suggested (see line 69).

16. Methods:

Design: please state what phase of study this was, i.e. phase II or III, pilot study etc

Response: We added this information (a randomized pilot and feasibility trial) and revised the title following the CONSORT guidelines (Eldridge et al., 2016)

17. Line 98: Please state what BIR stands for clearly, whether this is an inpatient setting, and the country of the study

Response: We clarified it as suggested (see line 134).

18. Participants: it would be clearer if you combined the inclusion criteria for the BIR programme with the inclusion criteria for the study

Response: We combined them as suggested (lines 138 – 148)

19. At what time point were participants assessed as meeting the inclusion criteria with the KF-NAP, was this in addition to the baseline assessment or is this done on admission?

Response: No, this was not in addition to the baseline assessment, which was described in the previous and current manuscripts (item f of the inclusion criteria). CBS via KF-NAP and other outcomes were assessed at baseline (T1). In other words, all the potentially eligibly patients were assessed at T1, if they met the neglect severity criterion (CBS > 10), then they were included in the study. If not, they were excluded from the study. 

20. Line 113: please clarify that the outcome assessors were different people to those who delivered the intervention

Response: We clarified this in the Outcome measures section as suggested (see line 200).

21. Line 130: please state how long each PAT session lasted.

Response: We added the information as suggested (see line 167).

22. Treatment: please state who delivered the treatment and whether it was always delivered as per the protocol.

Response: We added the information as suggested. The treatment was delivered by a physiotherapist, and it was always delivered as per the protocol. (see lines 162 and 179 – 180)

23. Outcome measures: please confirm whether the KF-NAP was carried out as per the assessment instructions i.e. same time of day each time, same location, all in one session. …. 

Response: We added the information as suggested (see lines 203 - 205).

24. Analysis: please state whether analysis was completed on an intention to treat basis

Response: As described in the previous and current manuscripts, in the Participant characteristics section of the Results, participants were included in the analyses if they completed the study. Thus, the results were not derived from the intent-to-treat basis. 

25. Results:

The baseline data presented in table 2 is quite confusing, it is not clear what ‘start’ means or how it is relevant, or why ego/allocentric is reported for scene copying but not for the bells test. Actual summary test scores would be more useful.

Response: Due to the space limit, we abbreviated words and phrases showing in the table. As shown in the notes under/after the table, “Start” stands for “starting column”, which was described in detail in the Methods section how the starting column indicates the presence of spatial neglect in the Bells test. The purpose of this table is to demonstrate whether a patient showed spatial neglect in different tests based on different tests’ own criteria. Showing different scores will be confusing as they do not immediately convey whether patients manifested neglect symptoms. 

26. It would be helpful to include all relevant statistics e.g. means, medians etc in a table as well as the graphs in order to aid interpretation for both hypothesis. This is required for journal policy.

Response: We have provided relevant stats when appropriate and necessary, while not repeating the text content, in the Tables. Figures 2 and 3 are already busy, and adding stats would make it visually confusing without improving the clarify of the results. We appreciated the comment and would like to discuss with the editors how to meet such requirements properly and at the same time make sure we are not repeating the information provided in the text. 

27. Discussion:

Line 337: Ten Brink used 10 dioptre prisms but you used 20 dioptre, this needs acknowledging

Response: Ten Brink et al. did not report the diopter of prism lenses used in their study. They used lenses that “[induce] an ipsilesional optical shift of 10°” (pp. 1019, Ten Brink et al., 2017). This suggests that they used lenses of 17.5 diopter. Thus, as reviewed in the Introduction, Ten Brink et al.’s study used stronger, rather than weaker, diopter prism lenses. 

28. Your point about all the different interventions patients received is interesting – the Cochrane review of spatial neglect covers all these types of treatment so it sounds like a bit of everything is best. Do you have a breakdown of types of other interventions patients received e.g. amount of limb activation etc?

Response: Yes, in fact it is a bit of everything as it’s a part of a general rehabilitation approach to patients in the BIR Program with lateralized deficits. 

29. You conclude that more study into PAT is needed – it would be interesting to test the neglect outcomes of patients in your programme generally to see if intensive combined treatment is the most effective treatment and perhaps research into PAT as a treatment for neglect alone needs to stop. Just a general thought based on your interesting findings.

Response: Thank you very much for this comment and suggestion. PAT has become a standard treatment for spatial neglect in the BIR Program since the study was completed. That is, the BIR is now even more comprehensive than before, adding PAT to therapists’ toolbox.

Reviewer #3: 

30. 1) Abstract: please specify the comparison intervention (vs. sham PAT) and population (patients with moderate-severe post-stroke SN).

Response: Thank you, information was added as suggested (see lines 30 and 32 - 33).

31. 2) Introduction: Similar comment to above, please state a clear PICO objective. e.g. The objective was to estimate the extent of the efficacy of a PAT program (nature, frequency, duration) vs. sham PAT program (xx) in improving xx among individuals with mod-sev post-stroke SN.

Response: Thank you, the objective was added to Introduction section as suggested (see lines 114 - 117).

32. 3) P7-8 missing references for CBS and Scene Copying Test.

Response: References added.

33. 4) Treatment: Please restate frequency and duration clearly in this paragraph. Add a picture of treatment setup (maybe as an appendix) and the sheets that were used for visual targets (was it computerized?).

Response: Frequency and duration were clarified as suggested (see lines 167 and 177) Treatment setup was added to Supporting information.

34. 5) Results: please start by how many were recruited and randomized, referring to your flow chart and only then proceed to discuss those what were analyzed. Also Table 1 must include info on all initially recruited 34 patients who were randomized (not sure why randomizing if a patient is not meeting your criteria re lesion location) and not only those who were included in the final analysis.

Response: We revised the paragraph as suggested (see lines 266 – 268) While we prefer to keep Table 1 as it is to demonstrate patients included in the final analyses, we added a table in the Appendix that includes all the initially recruited 34 patients. 

35. 6) Specify that your analysis was NOT intention-to-treat since you did not include those who did not finish the program in your analysis. It was per protocol.

Response: As described in the previous and current manuscripts, in the Participant characteristics section of the Results, participants were included in the analyses if they completed the study. Thus, the results were not derived from the intent-to-treat basis. 

36. 7) Important information for clinicians looking into adopting this treatment is who is the supplier for the prisms that were used. Can you indicate please. Also, can the treatment setup be accessed (e.g. existing computer program or existing treatment sheets that can be accessed by clinicians).

Response: The provider information was added in the Treatment section (see reference, line 163).

37. I wonder if another factor that have led to similar results in both groups is that your patients were mostly in acute/subacute phase of stroke recovery - where the window for recovery is of course larger. This can be one of your justifications and also a rationale to replicate such a study with chronic stroke patients (where we need more work as SN is known to persist in chronic phases of stroke recovery).

Response: We agreed that more work needs to be done in chronic phases of stroke recovery, which however, is not the focus or priority of the present study. The fact that we found both PAT and Sham groups improved to a similar extent cannot be explained by the phase of recovery given the study design.

---

## [Editor Report · Decision Letter 1]

4 Jan 2021

Prism adaptation treatment to address spatial neglect in an intensive rehabilitation program: A randomized pilot and feasibility trial

PONE-D-20-30567R1

Dear Dr. Vilimovský,

We’re pleased to inform you that your manuscript has been judged scientifically suitable for publication and will be formally accepted for publication once it meets all outstanding technical requirements. If possible, please make the raw data underlying the plots in Figures 2 and 3 publicly available either as a supplementary table or by posting them in a repository such as OSF.

Kind regards,

Daniel Mirman

Academic Editor

PLOS ONE
---

## [Editor Report · Acceptance letter]

7 Jan 2021

PONE-D-20-30567R1 

Prism adaptation treatment to address spatial neglect in an intensive rehabilitation program: A randomized pilot and feasibility trial 

Dear Dr. Vilimovský:

I'm pleased to inform you that your manuscript has been deemed suitable for publication in PLOS ONE. Congratulations! Your manuscript is now with our production department. 

Kind regards, 

on behalf of

Dr. Daniel Mirman 

Academic Editor

PLOS ONE